# The Role of Aquaculture in Shaping the Morphology of *Babylonia areolata*: A Comparative Study of Cultured and Wild Populations

**DOI:** 10.3390/biology14010039

**Published:** 2025-01-07

**Authors:** Haishan Wang, Zhi Chen, Yuhe Tong, Le Ye, Youming Li

**Affiliations:** 1Yazhou Bay Innovation Institute, Hainan Tropical Ocean University, Sanya 572022, China; jazz-123@163.com (H.W.);; 2Key Laboratory of Utilization and Conservation for Tropical Marine Bioresources, Ministry of Education, Hainan Tropical Ocean University, Sanya 572022, China

**Keywords:** *Babylonia areolata*, morphological variance analysis, multivariate analysis, aquaculture, morphological traits

## Abstract

Aquaculture plays an essential role in the global economy, and understanding how farming environments affect marine species is crucial for better resource management. This study focuses on *Babylonia areolata*, a type of marine mollusk, and compares the differences in their physical characteristics between wild and cultured populations. The main objective of this research was to examine how aquaculture environments influence the shape and size of these mollusks, which can impact their ability to adapt to natural conditions and their value in conservation. Our results showed distinct differences in the morphology of the mollusks from wild and cultured habitats, which have important implications for both aquaculture practices and the conservation of natural resources. The findings of this study could help improve aquaculture management strategies and contribute to the sustainable use of marine resources, benefiting both the industry and the environment.

## 1. Introduction

Mollusks are vital to aquaculture and marine ecosystems, serving as both a significant economic resource and an ecological component. The study of morphological characteristics is crucial for evaluating the success of aquaculture practices and preserving wild populations [1]. With the rapid expansion of aquaculture, understanding the morphological differences between cultured and wild populations has become increasingly important. These differences, shaped by environmental factors, also reflect genetic diversity and ecological adaptation [2,3].

*Babylonia areolata*, a commercially valuable marine mollusk, is widely cultured in Southeast Asia [4,5,6]. Overfishing and habitat destruction have significantly depleted wild populations of *B. areolata*, particularly in regions with high market demand. Unregulated harvesting practices disrupt population dynamics, reducing both the size and reproductive capacity of natural stocks [7,8]. The depletion of wild populations has made aquaculture the primary source of supply [9]. However, high-density rearing, monoculture feeding, and selective breeding in aquaculture environments induce morphological changes that can affect the adaptive capabilities, ecological roles, and market value of these mollusks [10]. Understanding the morphological differences between cultured and wild populations is therefore essential for optimizing aquaculture management and conserving wild resources.

Previous studies have revealed significant morphological differences between cultured and wild mollusks, particularly in shell shape, thickness, and body size. Studies have shown that environmental and rearing conditions significantly influence morphological traits in mollusks. For example, cultured *Mytilus chilensis* exhibited shorter and wider shells compared to their wild counterparts, a difference attributed to high-density rearing and uneven resource allocation [11]. Similarly, significant changes in shell shape and thickness were observed in *B. formosae habei* under high-density conditions [12]. More recent research has highlighted the combined role of genetic variability and environmental pressures in shaping the morphology of Mediterranean bivalves, emphasizing the importance of ecological and evolutionary factors in aquaculture [13,14].

This study aims to investigate the morphological differences between cultured and wild populations of *B. areolata* and to identify the environmental and genetic drivers of these differences. Using multivariate statistical analyses, including PCA, ANOVA, and LDA, this research provides a comprehensive understanding of morphological adaptations. Unlike previous studies that focused primarily on single traits, this study integrates multiple traits to reveal the complex interactions shaping mollusk morphology. The findings offer practical insights for optimizing aquaculture practices and conserving wild populations.

## 2. Materials and Methods

### 2.1. Study Area and Sampling

*B. areolata* samples were collected from both wild and cultured populations in Wanning, Hainan Province, China, a region known for its extensive aquaculture activities.

The wild population (64 individuals) was sampled through diving at Wuchang coastal waters (110°28′2.80″ E, 18°42′7.29″ N), located near Dong’ao Town, the primary aquaculture area for *B. areolata*. This coastal area is characterized by sandy substrates, moderate tidal fluctuations, and a diverse benthic ecosystem. During sampling, environmental parameters were recorded: the salinity was 33, water temperature ranged from 23 °C to 26 °C, and the density of the wild population was 3–7 individuals per square meter.

The cultured population (56 individuals) was obtained from Hainan Hongfuyang Aquatic Co., Ltd. (Sanya, China), a commercial aquaculture facility in the same region. These mollusks were reared in high-density tanks under controlled conditions. The aquaculture environment was maintained at a salinity of 31, with a water temperature of 27 ± 0.5 °C, and the stocking density was approximately 120 individuals per square meter.

### 2.2. Morphological Measurements

Nine morphological traits were measured for each individual using an electronic digital caliper with a precision of 0.01 mm (Figure 1).

Shell Length (SL): The longest distance from the apex to the edge of the shell base.

Shell Height (SH): Maximum vertical distance from the apex to the shell’s basal plane.

Body Spiral Height (BSH): Height of the largest body whorl from its base to the shell aperture.

Shell Aperture Length (SAL): Linear distance of the aperture opening from the base to the top.

Shell Aperture Width (SAW): Maximum horizontal width of the aperture.

Anterior Siphonal Canal Depth (FGAD): Vertical depth of the siphonal canal.

Anterior Siphonal Canal Width (FGAW): Maximum width of the siphonal canal.

Shell Thickness (ST): Average thickness of the shell measured at three points using a micrometer.

Shell Width (SW): Maximum width perpendicular to the shell’s length axis.

Three weight-related traits (total weight, shell weight, and fresh weight) were measured using an electronic balance with a precision of ±0.01 g. Measurements were repeated three times for each specimen, and mean values were used for analysis to minimize error.

### 2.3. Statistical Analysis

All morphological data were analyzed using multivariate statistical methods:

Principal Component Analysis (PCA): Used to reduce dimensionality and identify key traits driving population differences [15,16].

Linear Discriminant Analysis (LDA): Used to classify and differentiate between wild and cultured populations [17].

Univariate Analysis: ANOVA and Mann–Whitney U tests were employed to test for significant differences in individual traits [18].

Statistical analyses were conducted using Python (scikit-learn library, version 1.0.2) and R software (version 4.2.1), with a significance threshold of *p* < 0.05.

### 2.4. Ethical Approval

Ethical approval was not required for this study as it involved non-invasive sampling of marine mollusks, which are not protected under the relevant legislation in China.

## 3. Results

### 3.1. Morphological Parameter Comparison

Cultured populations exhibited significantly higher mean values in shell length, shell height, and total weight compared to wild populations (*p* < 0.05). For example, the shell length of cultured populations was 39.31 ± 2.41 mm, compared to 36.48 ± 3.77 mm in wild populations. The coefficient of variation (CV) was notably lower in the cultured group for traits like shell height (6.01% vs. 10.73%) and body spiral height (9.97% vs. 13.85%), indicating greater morphological uniformity in aquaculture environments. Wild populations displayed higher variability, reflecting the influence of environmental heterogeneity (Table 1).

### 3.2. Correlation Analysis

Spearman correlation coefficients were calculated to assess the relationships between morphological traits in wild and cultured populations of *B. areolata*. To provide a more intuitive understanding, scatter plots and heatmaps were generated to visualize the linear trends and pairwise correlations between traits.

The correlation heatmap and scatter plot matrix for the wild population. In the wild population, most morphological traits displayed strong positive correlations, particularly among shell length (SL), shell height (SH), and shell width (SW), which were highly correlated with weight-related traits (correlation coefficients > 0.9). These patterns suggest well-coordinated growth in natural environments. Conversely, traits like shell thickness (ST) and siphonal canal dimensions (FGAW, FGAD) showed weaker correlations, reflecting their independence from overall growth patterns. The findings highlight the selective pressures in natural environments that drive adaptive coordination among key traits (Figure 2).

The correlation heatmap and scatter plot matrix for the cultured population. In the cultured population, correlations among traits were generally weaker than in the wild population, with moderate correlations observed for shell length (SL) and shell height (SH) (correlation coefficients ~0.8). Traits such as shell thickness (ST) exhibited slight negative correlations with body spiral height (BSH) and siphonal canal dimensions (FGAW, FGAD), suggesting independent development under controlled conditions. Weight-related traits, including total weight (TW), shell weight (SWt), and fresh weight (FW), also displayed reduced correlations with other traits. These findings reflect the uniformity of aquaculture environments, which likely limit the development of morphological variability (Figure 3).

### 3.3. Standardized Data Descriptive Statistics

To minimize the impact of individual size differences, standardized morphological parameters were analyzed for both wild and cultivated groups (Table 2). The results revealed significant differences in several traits, reflecting the influence of distinct growth environments on morphological development.

The cultivated group exhibited higher mean values for traits such as relative shell height (R_SH: 0.5924 ± 0.0236 vs. 0.5753 ± 0.0233) and relative fresh weight (R_FW: 0.1616 ± 0.0377 vs. 0.1050 ± 0.0256), indicating that controlled conditions promote enhanced growth. However, certain traits showed greater variability in the cultivated group, such as R_FW, with a higher standard deviation (0.0377 vs. 0.0256), suggesting that uniform cultivation practices may not fully standardize growth outcomes.

In contrast, the wild group exhibited lower mean values but higher variability for traits like relative total weight (R_TW: CV = 20.55% vs. 15.53%) and relative shell weight (R_SWt: CV = 21.06% vs. 13.11%), reflecting the impact of environmental heterogeneity and resource variability in natural habitats.

These findings highlight how controlled aquaculture conditions enhance growth efficiency but may increase variability for some traits. In contrast, the higher variability observed in wild populations underscores their adaptive responses to fluctuating environmental pressures. Introducing selective pressures in cultivation could help minimize variability and enhance consistency in morphological traits.

### 3.4. Significant Differences in Morphological Parameters

The violin plot analysis illustrates significant differences in standardized morphological parameters between the wild and cultivated groups of *B. areolata*. Statistical significance was determined using ANOVA for normally distributed data and the Mann–Whitney U test for non-normally distributed data (Figure 4).

Several traits showed significant differences (*p* < 0.05), highlighting the distinct influences of cultivation and wild environments:

Relative Shell Height (R_SH): Higher in the cultivated group, reflecting consistent growth conditions in aquaculture. Relative Total Weight (R_TW) and Relative Fresh Weight (R_FW): Both significantly greater in the cultivated group, emphasizing the positive impact of stable nutrient availability in aquaculture. Relative Shell Thickness (R_ST): Thicker shells in the wild group suggest adaptations to natural environmental factors such as predation and variable calcium levels.

In contrast, some parameters showed no significant differences (*p* > 0.05):

Relative Body Spiral Height (R_BSH): Similar across groups, indicating limited environmental influence. Relative Shell Aperture Width (R_FGAW) and Relative Shell Weight (R_SWt): No significant differences, suggesting shared developmental patterns in both environments.

These findings underscore how controlled aquaculture environments promote uniform growth in traits like shell height and weight, while wild environments drive greater variability and adaptations in traits such as shell thickness. Significant traits, such as R_SH, R_TW, and R_FW, serve as reference points for understanding environmental impacts on morphology, whereas non-significant traits highlight inherent stability across both environments.

### 3.5. Results and Analysis of Principal Component Analysis (PCA)

Principal Component Analysis (PCA) was applied to standardized morphological data to identify key differences between the wild and cultivated groups of *B. areolata*. The PCA scatterplot shows the separation of the two groups (Figure 5), while the loading plots highlight the contributions of individual traits to the principal components (Figure 6).

PCA1 and PCA2 together accounted for 91.96% of the total variance, indicating the effectiveness of PCA in capturing the main patterns in morphological variation (Figure 5). The PCA scatterplot clearly shows a distinct separation between the wild (red) and cultivated (green) groups along PCA1. This suggests that environmental factors in cultivation and wild settings drive significant differences in overall shell morphology. The limited overlap along PCA2 indicates that some secondary traits are shared between the groups, possibly due to overlapping environmental or genetic factors.

The PCA loading plots illustrate the contributions of standardized morphological traits to PCA1 and PCA2 in both wild and cultivated groups (Figure 6).

Wild Group: PCA1 was mainly influenced by relative shell thickness (R_ST), relative shell height (R_SH), and relative shell length (R_SL), reflecting size and robustness critical for adaptation in natural environments. PCA2 was driven by relative anterior siphonal canal width (R_FGAW) and relative shell aperture width (R_SAW), highlighting ecological adaptations for survival in variable conditions.

Cultivated Group: PCA1 was dominated by relative shell thickness (R_ST), relative shell width (R_SW), and relative shell height (R_SH), emphasizing growth uniformity in controlled environments. PCA2 was shaped by relative fresh weight (R_FW) and relative shell weight (R_SWt), underscoring the influence of nutrient availability on growth dynamics.

These findings reveal how natural selection and resource heterogeneity in wild environments promote adaptive traits, while controlled aquaculture conditions reduce selective pressures and prioritize uniform growth. Traits like R_ST and R_FW are critical for optimizing aquaculture practices, offering pathways to enhance desirable traits while maintaining morphological diversity.

### 3.6. Discriminant Analysis Results

Linear Discriminant Analysis (LDA) was conducted on standardized morphological data to identify key traits differentiating wild and cultivated populations of *B. areolata*. The results showed a distinct separation between the two groups along the LDA1 axis, achieving high classification accuracy and underscoring the robustness of morphological traits in distinguishing environmental influences.

The LDA model achieved a classification accuracy of 94.44%, effectively distinguishing wild and cultivated populations of *B. areolata* based on morphological traits (Figure 7). The frequency distribution along the LDA1 axis showed clear separation, with the wild population predominantly clustering in the negative range and the cultivated population in the positive range. This separation reflects significant morphological differences driven by distinct environmental and growth conditions.

Key traits contributing to the separation included relative shell thickness (R_ST), relative fresh weight (R_FW), and relative shell aperture length (R_SAL). R_ST was a major driver, reflecting the influence of predation and resource variability in the wild group versus resource optimization in the cultivated group. R_FW, significantly higher in the cultivated group, emphasized the role of controlled nutrient availability, while R_SAL highlighted differences in shell development patterns influenced by growth environments.

The high classification accuracy confirms the reliability of morphological traits as robust indicators for distinguishing populations. These findings provide a foundation for further research to understand the drivers of morphological variation and to optimize aquaculture practices, ensuring desirable traits are maintained while promoting sustainable management strategies.

## 4. Discussion

### 4.1. The Influence of Environment on Morphological Development

Environmental conditions, particularly stability and heterogeneity, directly shape the morphological development of *B. areolata*. This study revealed significant morphological differences between wild and cultivated populations, particularly in traits such as shell height, shell length, total weight, and aperture length.

In aquaculture environments, mollusks live under stable conditions, including controlled water quality, consistent nutrient supply, and reduced predation pressure. These factors promote morphological uniformity, with cultivated populations exhibiting larger shell dimensions and total weights. Consistent with previous studies, stable aquaculture conditions enhance growth efficiency and trait consistency [19,20,21]. For instance, research on Mediterranean bivalves has shown that stable environmental conditions combined with limited genetic variability promote uniform morphology [13,20].

In contrast, wild populations inhabit heterogeneous environments characterized by fluctuating tides, variable salinity, and patchy benthic resources. Such conditions create selective pressures that drive morphological diversity. For instance, thicker shells may offer better protection against predation, while variations in aperture size optimize energy allocation under resource constraints [22]. Predation pressure and resource variability in wild environments drive the evolution of defensive morphological traits in mollusks [23]. Moreover, morphological diversity in bivalves and gastropods reflects adaptive responses to heterogeneous habitats [24]. 

This study further underscores the contrasting impacts of stable aquaculture environments and fluctuating wild habitats on morphological traits, emphasizing the significant role of environmental conditions in shaping morphological diversity [24].

### 4.2. The Role of Genetic and Environmental Interactions

The interaction between genetic diversity and environmental conditions plays a crucial role in shaping the morphological differences between wild and cultivated populations. This study demonstrates that both environmental conditions and genetic backgrounds jointly influence the adaptive traits of *B. areolata*.

In aquaculture environments, stable conditions and selective breeding reduce selective pressures, leading to decreased genetic diversity. This genetic homogeneity promotes morphological uniformity but limits adaptability, increasing vulnerability to stressors such as disease outbreaks and climate fluctuations [25,26].

In contrast, wild populations exhibit greater genetic diversity, allowing them to adapt more effectively to complex environmental pressures. Traits such as aperture length and shell thickness reflect adaptive responses to predation risks and resource variability [23,27]. Studies on mollusks have shown that genetic variation plays a critical role in enabling populations to respond to dynamic environments [28,29], with synergistic effects between genetic diversity and environmental conditions shaping morphological traits [30].

This study corroborates these findings, highlighting how resource stability and artificial selection in aquaculture lead to morphological uniformity, while environmental variability and predation pressures in the wild promote greater morphological and genetic diversity. Defensive traits, such as shell thickening, are particularly linked to genetic variability [23,31]. As noted, these traits are more pronounced in populations facing greater environmental challenges.

Overall, the synergy between genetic diversity and environmental factors shapes adaptive traits, enabling wild populations to thrive in complex ecological conditions [32], while cultivated populations reflect the constraints of reduced genetic diversity under artificial selective pressures [31].

### 4.3. Optimization of Aquaculture Management

Balancing growth efficiency and genetic diversity is crucial for sustainable aquaculture management. While aquaculture systems enhance growth and promote morphological uniformity, they often reduce genetic diversity, which limits adaptability to environmental changes and increases vulnerability to stressors like disease outbreaks or climate fluctuations [32].

To mitigate these risks, aquaculture practices should incorporate natural selection pressures and enhance genetic diversity. Strategies such as adjusting stocking densities, introducing diverse habitat structures, and interbreeding with wild populations can promote adaptive traits and morphological diversity [33]. Crossbreeding between strains and rotational breeding programs can also increase genetic variability, ensuring populations remain resilient to future challenges [34,35].

Recent research on *Mytilus* spp. has demonstrated the value of diverse environmental conditions and genetic interactions in fostering morphological robustness in cultivated populations [32]. Advanced genomic selection technologies further provide opportunities to optimize growth traits while maintaining genetic diversity [36]. These approaches offer a balanced pathway to achieving higher productivity alongside ecological sustainability, addressing both economic and environmental objectives in aquaculture systems [37].

## 5. Conclusions

This study demonstrates that aquaculture environments significantly influence the morphology of *B. areolata*, resulting in more uniform traits compared to wild populations. Cultivated populations exhibited consistent growth in traits such as shell length, shell height, and total weight under stable conditions, while wild populations showed higher variability in traits like shell thickness and aperture length, reflecting the impact of environmental heterogeneity and natural selection pressures.

The findings highlight the dual role of environmental and genetic factors in shaping mollusk morphology. Optimized aquaculture conditions enhance growth efficiency but may reduce genetic diversity, limiting adaptability to environmental changes. Conversely, wild populations benefit from greater genetic diversity, promoting resilience to fluctuating conditions. These results underscore the need to balance growth optimization with genetic conservation in aquaculture management.

Recommendations:

Simulating Natural Selection Pressures: Introduce variability in aquaculture environments, such as adjusting stocking densities and diversifying habitat structures, to mimic natural selection pressures and enhance the morphological adaptability of cultured populations.

Genetic Diversity Conservation: Implement crossbreeding programs between different cultured strains and interbreeding with wild populations to maintain genetic variability and improve population resilience.

Sustainable Management Practices: Promote sustainable fishing practices and establish marine protected areas to preserve the genetic and morphological diversity of wild *B. areolata* populations.

Future Research Directions: Investigate the combined effects of environmental pressures and genetic factors on mollusk morphology over multiple generations, considering the potential impacts of climate change on both wild and cultured populations.

## Figures and Tables

**Figure 1 biology-14-00039-f001:**
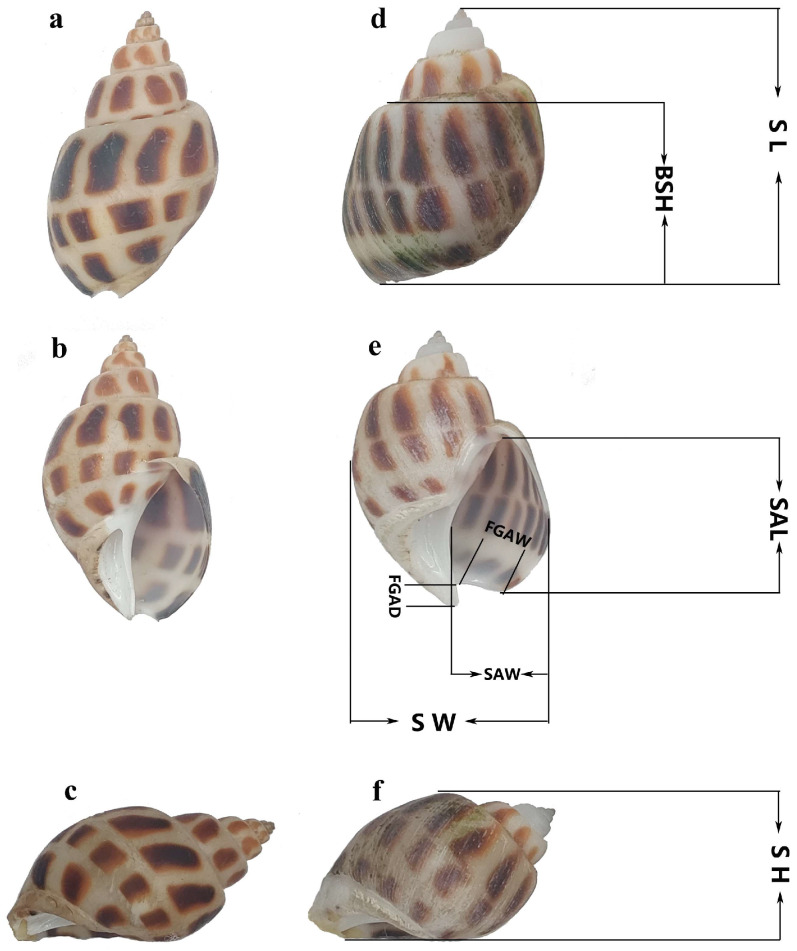
Morphological parameters and specimen photographs of *B. areolata*. This figure presents the morphological characteristics and measurements of wild and cultured *B. areolata* individuals. (**a**) Dorsal view of a wild individual; (**b**) ventral view of a wild individual; (**c**) lateral view of a wild individual; (**d**) dorsal view of a cultured individual; (**e**) ventral view of a cultured individual; (**f**) lateral view of a cultured individual. The labeled parameters include SL (Shell Length), SH (Shell Height), BSH (Body Spiral Height), SAL (Shell Aperture Length), SAW (Shell Aperture Width), SW (Shell Width), FGAD (Anterior Siphonal Canal Depth), and FGAW (Anterior Siphonal Canal Width). These parameters are used to assess the morphological differences between wild and cultured populations.

**Figure 2 biology-14-00039-f002:**
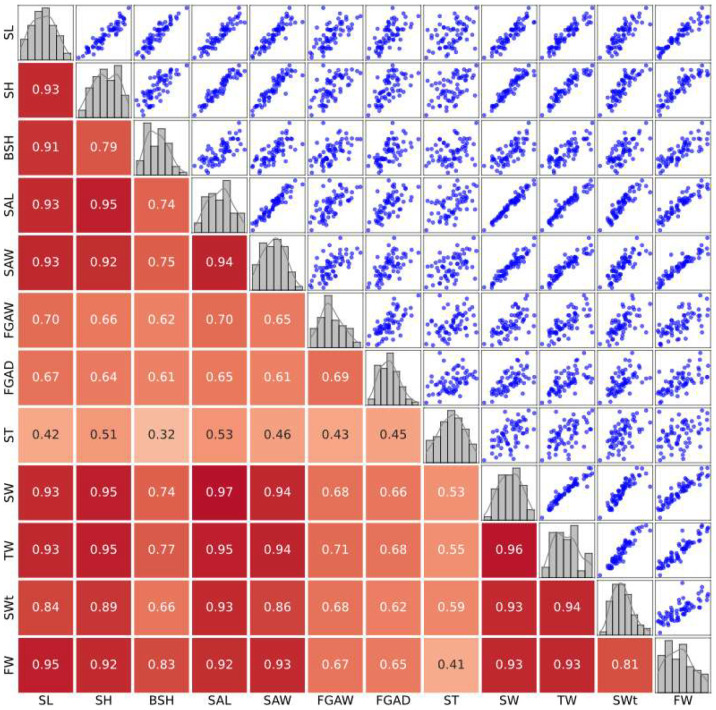
Correlation and scatter plot matrix for morphological traits in the wild population. The correlation heatmap and scatter plot matrix for the wild population of *B. areolata*. Histograms along the diagonal show trait distributions, scatter plots below the diagonal reveal pairwise relationships, and correlation coefficients are displayed above the diagonal. The colors in the heatmap represent the strength of the correlation: darker red indicates stronger positive correlations, while lighter shades indicate weaker correlations.

**Figure 3 biology-14-00039-f003:**
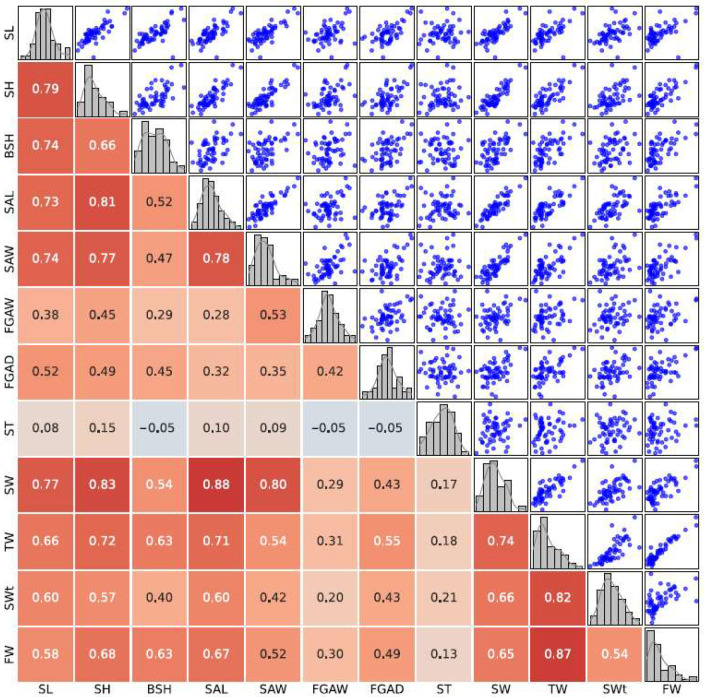
Correlation and scatter plot matrix for morphological traits in the cultured population. The correlation heatmap and scatter plot matrix for the cultured population of *B. areolata*. Histograms along the diagonal show trait distributions, scatter plots below the diagonal reveal pairwise relationships, and correlation coefficients are displayed above the diagonal. The colors in the heatmap represent the strength of the correlation: darker red indicates stronger positive correlations, while lighter shades indicate weaker correlations.

**Figure 4 biology-14-00039-f004:**
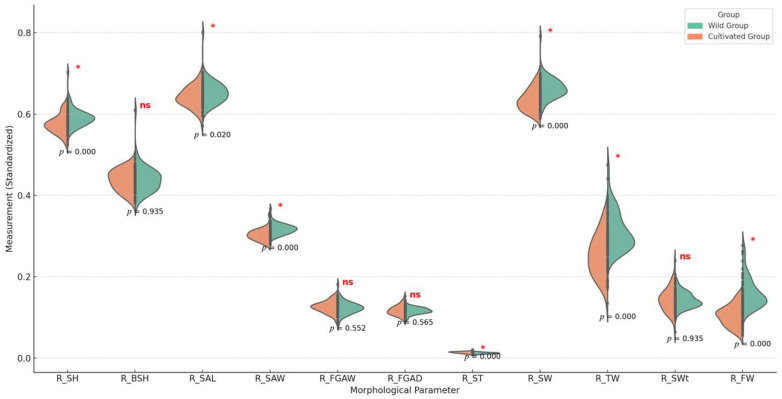
Violin plot of morphological parameters between groups. The violin plots compare standardized morphological parameters between the wild group (green) and the cultivated group (orange). Significant differences (*p* < 0.05) are marked with a red asterisk (*), while non-significant results are labeled as “ns”.

**Figure 5 biology-14-00039-f005:**
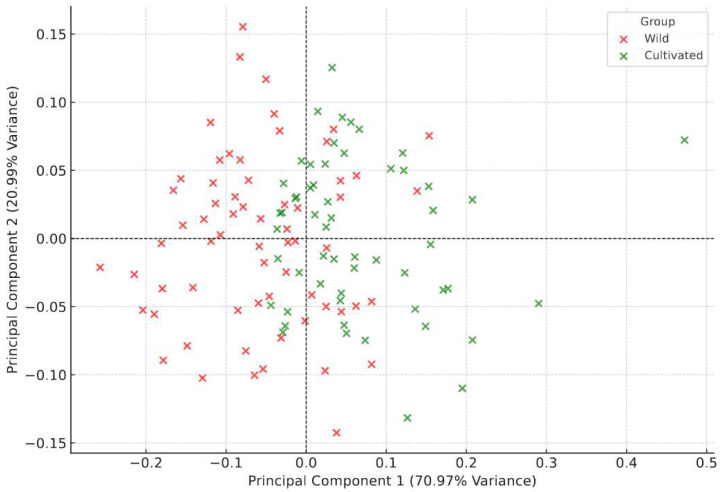
PCA plot of standardized morphological data. The PCA scatterplot illustrates the distribution of wild (red) and cultivated (green) groups along PCA1 and PCA2.

**Figure 6 biology-14-00039-f006:**
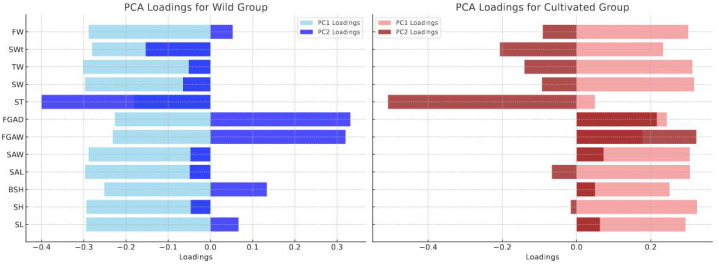
PCA loadings for wild and cultivated groups. The bar plots illustrate the contributions of morphological traits to PCA1 and PCA2 for the wild group (blue) and the cultivated group (red).

**Figure 7 biology-14-00039-f007:**
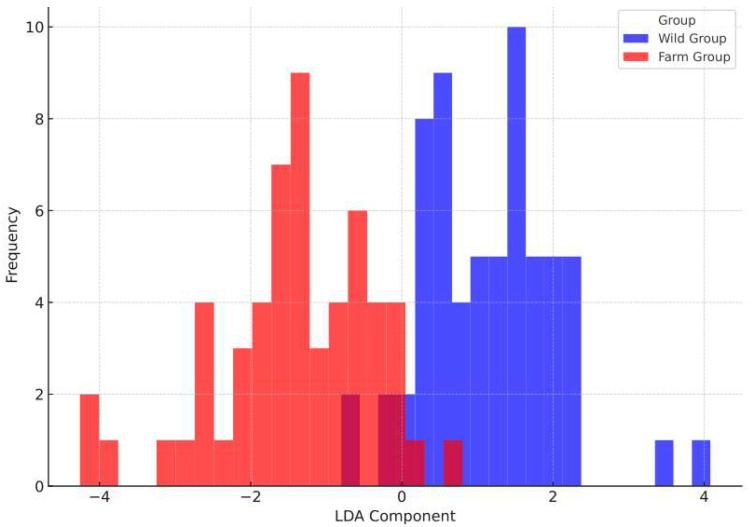
Linear discriminant analysis for wild and cultivated groups. The LDA projection plot displays the frequency distribution of wild (blue) and cultivated (red) groups along the LDA1 axis. The wild population clusters predominantly in the negative range, while the cultivated population clusters in the positive range, reflecting distinct morphological differences between the two groups. The dark red bars represent the overlapping region between the two groups.

**Table 1 biology-14-00039-t001:** Morphological characteristics of wild and cultivated groups of *B. areolata*.

Parameter	Wild Group		Cultivated Group	
	Mean ± SD	CV	Cultivated	CV
SL	36.48 ± 3.77	10.33%	39.31 ± 2.41	6.14%
SH	20.98 ± 2.25	10.73%	23.27 ± 1.40	6.01%
BSH	15.98 ± 2.21	13.85%	17.32 ± 1.73	9.97%
SAL	23.40 ± 2.46	10.52%	25.69 ± 1.24	4.84%
SAW	11.09 ± 1.30	11.75%	12.49 ± 0.75	6.03%
FGAW	4.58 ± 0.75	16.29%	4.87 ± 0.67	13.86%
FGAD	4.28 ± 0.61	14.24%	4.57 ± 0.39	8.58%
ST	0.51 ± 0.09	17.74%	0.49 ± 0.07	14.41%
SW	23.14 ± 2.40	10.39%	26.10 ± 1.39	5.34%
TW	9.43 ± 2.79	29.56%	12.39 ± 2.31	18.66%
SWt	5.30 ± 1.52	28.73%	5.63 ± 0.84	14.93%
FW	3.91 ± 1.30	33.19%	6.39 ± 1.72	26.96%

**Table 2 biology-14-00039-t002:** Standardized morphological parameters in wild and cultivated groups.

Parameter	Wild Group		Cultivated Group	
	Mean ± SD	CV	Cultivated	CV
R_BSH	0.4371 ± 0.0261	5.98%	0.4405 ± 0.0345	7.83%
R_FGAD	0.1175 ± 0.0122	10.36%	0.1164 ± 0.0080	6.91%
R_FGAW	0.1256 ± 0.0145	11.51%	0.1239 ± 0.0161	12.98%
R_FW	0.1050 ± 0.0256	24.37%	0.1616 ± 0.0377	23.32%
R_SAL	0.6419 ± 0.0260	4.05%	0.6545 ± 0.0297	4.53%
R_SAW	0.3038 ± 0.0150	4.92%	0.3180 ± 0.0130	4.09%
R_SH	0.5753 ± 0.0233	4.04%	0.5924 ± 0.0236	3.99%
R_ST	0.0140 ± 0.0023	16.47%	0.0125 ± 0.0019	15.14%
R_SW	0.6349 ± 0.0268	4.23%	0.6647 ± 0.0279	4.19%
R_SWt	0.1434 ± 0.0302	21.06%	0.1431 ± 0.0188	13.11%
R_TW	0.2539 ± 0.0522	20.55%	0.3142 ± 0.0488	15.53%

## Data Availability

Dataset available upon request from the authors.

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
