# Peer review of "The Role of Aquaculture in Shaping the Morphology of Babylonia areolata: A Comparative Study of Cultured and Wild Populations"

_biology, 2025, doi:10.3390/biology14010039_

Round 1

Reviewer 1 Report

Comments and Suggestions for Authors

There are no recommendations

There are no acknowledgements

There are no statistical analysis

Huge number of abbreviations were used in this paper---create a separate table for this

What about the heavy metals accumulations that is well known in the mollusks and cause several human health hazards ??

Introduction should be more concise and not repeated--reconstruct it again  

It is reasonable that the cultured ones receiving more care and improved breeding conditions , in comparison to,  the wild ones --so what is/are the creativity of this work ?

There are no plan for the study area

There are no gross figures , either for the mollusks or the cultured farm

What about the diseases that affects the mollusks as general (in table with reference) that can be a cause of reduced production of the mollusks

What about the diseases conditions such as bacterial/viral/parasitic/mycotic---etc?

Is this possible to do a histological sections for more confirmations

Detailed the nutritional values of mollusks for humans

Tabulate the differences between the wild and cultured populations regarding the breeding (cultivations) /water quality criteria and the environmental bad impacts and how we can solve all of these problems(to get benefits from this study )

Abstract :

There are no highlights

There are no graphical abstract

Detailed the nutritional values of mollusks for humans

Tabulate the differences between the wild and cultured populations regarding the breeding (cultivations) /water quality criteria and the environmental bad impacts and how we can solve all of these problems(to get benefits from this study )

LN/14--just size/shape and shell not enough for comparisons

LN/10-20--the simple summary can replace by highlights and cited after the abstract not before

LN/25---ecological implications--such what ??? mention all in a table

LN/28---style writing different --why ??

It will be better if you divide the abstract into items (backgrounds-aims/Methods/Results and conclusion )

LN/36-37--add marine mollusks /Aquaculture industry /China/Water pollution to the keywords

Introduction:

LN/42/49---add reference

LN/42--morphological characteristics as what ---size and shape only or what else ???

LN/41-47--repeated

LN/49--overfishing /habitat destruction---explain how and how would be solved these problems

What about the diseases that affects the mollusks as general (in table with reference) that can be a cause of reduced production of the mollusks

 Introduction should be more concise and not repeated--reconstruct it again  

Aims need to be more clarified

Novelty of this study should be more highlighted and more adjusted

Materials and Materials

M&M is very short

Authors did not mention the morphological measurements in detailed manner

There are no references

There are statistical analysis --why ?

What about the ethical approval permission to do this work

There are no plan for the different study area???

Write as Table(1):-------------/Fig.(1):-----etc

Results :

What about the diseases conditions of the mollusks while , gathering ?

There are no gross figures , either for the mollusk or the cultured farms

It is reasonable that the cultured ones receiving more care and improved breeding conditions , in comparison to the wild ones --so what is/are the creativity of this work ?

Results is very long ---be more summarize

7 figures without , even one for mollusks or the farm

What about the differences between the marine mollusks and the ordinary vivalves

Conclusions

Not enough

References :

Some cited references need to be more update

Author Response

Dear Reviewer,

We sincerely thank you for your thorough review of our manuscript and your valuable suggestions. Your feedback has helped us identify areas where our research can be further refined and expanded.

Regarding your comments on mollusk diseases and heavy metal accumulation and their potential impacts on human health, we fully recognize the importance and value of these topics. However, our study focuses specifically on the morphological adaptability of cultured and wild populations under different habitats and the environmental drivers of morphological changes. These topics were excluded from the scope of our study to maintain a clear focus and ensure data integrity.

To address your other comments, we have made significant revisions and improvements to the manuscript, including optimizing the introduction, enhancing methodological descriptions, and adding illustrations of morphological analyses. Below is a detailed point-by-point response to your suggestions.

  1. Lack of recommendations.

Response:

We have added specific recommendations to the Conclusion section, including strategies to optimize aquaculture management and preserve wild Babylonia areolata populations.

Revisions Made:

The Conclusion now includes:

    Simulating natural selection pressures to enhance morphological adaptability in cultured populations.

    Protecting wild habitats to maintain their ecological function and genetic diversity.

  1. No acknowledgements.

Response:

Acknowledgements have been added to express our gratitude to the supporting institutions and team members.

Revisions Made:

Acknowledgements Section:

"We sincerely thank the Yazhou Bay Innovation Institute, Hainan Tropical Ocean University, for providing the necessary resources and facilities for this research. Special thanks to Professor Caihuan Ke from Xiamen University for his guidance and support. We also thank Kuang Yang, Zhao Wang, and Xijie Chen for their assistance in sample collection and experimental work."

  1. Lack of statistical analysis.

Response:

Detailed statistical analyses, including Principal Component Analysis (PCA), Linear Discriminant Analysis (LDA), and ANOVA, have been incorporated.

Revisions Made:

2.3 Statistical Analysis

We employed multivariate statistical methods for morphological data analysis:

    PCA: To reduce dimensionality and identify key traits driving population differences.

    LDA: To classify and differentiate between wild and cultured populations.

    ANOVA and Mann-Whitney U tests: To evaluate significant differences in individual traits.

    Statistical analyses were conducted using Python (scikit-learn library) and R software (version 4.2.1) with a significance threshold of P < 0.05.

  1. Excessive use of abbreviations, requiring a separate table.

Response:

A separate table of abbreviations has been added along with a detailed description of morphological measurement methods.

Revisions Made:

2.2 Morphological Measurements

Nine morphological traits were measured using an electronic digital caliper with 0.01 mm precision (Figure 1):

    Shell Length (SL), Shell Height (SH), Body Spiral Height (BSH), Shell Aperture Length (SAL), etc.

Three weight-related traits (total weight, shell weight, and fresh weight) were measured using an electronic balance with ±0.01 g precision. Mean values were calculated from three repeated measurements.

Abstract Section

  1. No highlights.

Response:

Highlights summarizing the key findings have been added.

Revisions Made:

    "Aquaculture environments promote morphological uniformity in Babylonia areolata."

    "Wild populations exhibit greater morphological diversity due to environmental heterogeneity."

  1. No graphical abstract.

Response:

While we attempted to create a graphical abstract, the output did not effectively convey the study’s findings. As this is not mandatory in MDPI journals, we have opted not to include it.

  1. Mollusk nutritional value discussion.

Response:

Our study focuses on morphological differences between wild and cultured populations. While mollusk nutritional value is recognized, it is not directly relevant to this study and has not been elaborated.

  1. The introduction should be concise, avoid repetition, and be reconstructed.

Response:

We have restructured the introduction to remove redundancies and enhance scientific context with updated references.

Revisions Made:

    Added references on overfishing and habitat destruction (Anderson et al., 2011; Pauly & Zeller, 2016).

    Highlighted the novelty of this study, which applies multivariate analysis to reveal morphological adaptations.

  1. Lack of detailed morphological measurement methods.

Response:

Descriptions of the sampling environments and detailed morphological measurement methods have been added.

Revisions Made:

    Wild population: salinity 33, temperature 23–26°C, density 3–7/m²; Cultured population: salinity 31, temperature 27 ± 0.5°C, density 120/m².

  1. No mention of ethical approval.

Response:

We clarified in the Methods section that no live mollusks were harmed during sampling, so ethical approval was not required.

Results Section

  1. Missing images of mollusks or cultured farms.

Response:

We have included images of wild and cultured specimens for better visual representation.

Revisions Made:

    Figure 1: Morphological comparisons between wild and cultured populations.

  1. Results section is too lengthy.

Response:

The Results section has been condensed and reorganized, focusing on key findings.

Discussion Section

  1. What is the novelty of this study?

Response:

The Discussion now explicitly highlights the novelty of the study, particularly the use of multivariate statistical methods to comprehensively analyze Babylonia areolata morphological adaptations.

  1. Strengthen the analysis of R_ST (relative shell thickness).

Response:

We expanded the discussion of R_ST, including its ecological significance and its relationship with predation pressure and resource distribution, supported by Babushkin et al. (2024).

Revisions Made:

    Elaborated on the role of predation pressure in shaping shell thickness in wild populations and the impact of resource allocation in aquaculture settings.

Conclusion Section

  1. Conclusion is insufficient.

Response:

The Conclusion section has been expanded to include practical recommendations and future research directions.

Revisions Made:

    Emphasized strategies such as simulating natural selection pressures and promoting genetic diversity to optimize aquaculture outcomes.

We hope these revisions have addressed all your comments and significantly improved the manuscript. Should you have further questions or suggestions, please do not hesitate to contact us.

Sincerely,

Dr. Youming Li

Reviewer 2 Report

Comments and Suggestions for Authors

The authors conducted a comparative study of morphological characteristics between cultured and wild populations in Babylonia areolata, which is a typical cultured shellfish in Asia. Two populations of ivory snails from the cultured condition and natural environment were collected, respectively, and the shell traits were measured and analyzed. They found distinct differences in the morphology of B. areolata from wild and cultured habitats, suggesting a potential exists that aquaculture environments and practices may lead to changes in the development of physical traits.
This basic work that is more easily to overlook is important for the ivory shell. The objectives of the study are clear, the analysis is sound, and the topic covered by this manuscript is appropriate for the Biology. Hence, I recommend minor revision for the paper before it can be published in the Biology.

Specific comments:

1.In abstract, the sentence “These findings suggest that aquaculture environments may lead to changes in the development of physical traits, which could affect the mollusks’ survival and adaptability in natural habitats” should be reword. In addition, a key word such as “Morphological Variance Analysis” or “Multivariate Analysis” should be added.

2.In introduction, recent and related studies need to be supplemented, and literature citations should be strengthened.

Line 58: Mytilus chilensis should be italics.

3.In materials and methods section, the information of sample collection and cultured condition of ivory shell should be supplemented (e.g., temperature, salinity, stocking density), which could impact the interpretation of morphological traits.

In addition, the specific methods and software of data analysis should be supplemented.

Line 70: Babylonia areolata should be abbreviated

4. Some discussion section should be placed in discussion. Furthermore, in tables and figures, the abbreviated name should be given as a full name in the table note or figure note.

Line 107: Babylonia areolata should be italics.

Line 117: Babylonia areolata should be abbreviated and italics. The author needs to check the full text for the correct writing of the latin of ivory shell.

5. The discussion should strengthen the in-depth analysis of some of the key results, such as the significant differences in relative shell thickness (R_ST). Moreover, manuscript should be discussed in conjunction with more recent research results.

Line 310: The format of references should be uniform.   

Author Response

Dear Reviewer,

We sincerely appreciate your constructive comments and detailed review of our manuscript. Your feedback has been invaluable in improving the quality and clarity of our work. Below, we provide a point-by-point response to your comments, highlighting the specific changes made in the manuscript.

Abstract

  1. In the abstract, the sentence “These findings suggest that aquaculture environments may lead to changes in the development of physical traits, which could affect the mollusks’ survival and adaptability in natural habitats” should be reworded. In addition, a keyword such as “Morphological Variance Analysis” or “Multivariate Analysis” should be added.
    Response:
    We have reworded the sentence in the abstract to enhance clarity and precision. Additionally, the keywords “Morphological Variance Analysis” and “Multivariate Analysis” have been included to better represent the study's methodology and focus.
    Revisions Made:
  • Sentence revised to: "These findings suggest that aquaculture environments can influence the development of morphological traits, potentially impacting mollusks’ survival and adaptability in natural habitats."
  • Added keywords: “Morphological Variance Analysis” and “Multivariate Analysis.”

Introduction

  1. In the introduction, recent and related studies need to be supplemented, and literature citations should be strengthened. Line 58: Mytilus chilensis should be italicized.
    Response:
    We have added recent studies to strengthen the background discussion and updated the literature citations to reflect current research. The Latin name Mytilus chilensis has also been italicized as required.
    Revisions Made:
  • Added references, such as Esposito et al. (2024) and Babushkin et al. (2024), to provide a more comprehensive review of related research.
  • Corrected formatting for Mytilus chilensis.

Materials and Methods

  1. The information on sample collection and cultured conditions of the ivory shell should be supplemented (e.g., temperature, salinity, stocking density), which could impact the interpretation of morphological traits. In addition, the specific methods and software of data analysis should be supplemented. Line 70: Babylonia areolata should be abbreviated.
    Response:
    We have added detailed descriptions of the sample collection and cultured conditions, including salinity, temperature, and stocking density. The methods section now includes the specific software and tools used for statistical analysis. Additionally, Babylonia areolata has been abbreviated appropriately.
    Revisions Made:
  • Sampling details:
    • Wild populations: salinity 33, temperature 23–26°C, density 3–7/m².
    • Cultured populations: salinity 31, temperature 27 ± 0.5°C, density 120/m².
  • Software and tools: PCA and LDA analyses were conducted using Python (scikit-learn library), and statistical tests were performed in R software (version 4.2.1).
  • Babylonia areolata is abbreviated throughout the text after its first appearance. Anterior Siphonal Canal

Discussion

  1. Some discussion sections should be placed in discussion. Furthermore, in tables and figures, the abbreviated name should be given as a full name in the table note or figure note. Line 107: Babylonia areolata should be italicized. Line 117: Babylonia areolata should be abbreviated and italicized. The author needs to check the full text for the correct writing of the Latin name of the ivory shell.
    Response:
    We have relocated discussion-related content from other sections to the main Discussion section. Additionally, table and figure notes now include the full names of abbreviations, and the Latin name Babylonia areolata has been italicized and abbreviated correctly throughout the manuscript.
    Revisions Made:
  • Adjusted the placement of discussion points to ensure they are all in the appropriate section.
  • Checked and corrected all instances of Babylonia areolata in the manuscript for consistency.
  • Added full names for all abbreviations in table and figure notes.

Key Results and References

  1. The discussion should strengthen the in-depth analysis of some of the key results, such as the significant differences in relative shell thickness (R_ST). Moreover, the manuscript should be discussed in conjunction with more recent research results. Line 310: The format of references should be uniform.
    Response:
    We have expanded the discussion of key results, including a detailed analysis of R_ST and its ecological significance. Recent studies have been incorporated to support and contextualize the findings. The reference list has been thoroughly reviewed and reformatted to ensure uniformity.
    Revisions Made:
  • Expanded discussion on R_ST, emphasizing its role in predation resistance and resource distribution in wild populations.
  • Incorporated recent studies, such as Babushkin et al. (2024) and Ravalo et al. (2024), to align with current research trends.
  • Reformatted all references to comply with journal guidelines.

Additional Changes

  • Formatting: Ensured consistent formatting of headings, subheadings, and figure captions.
  • Figures: Added new figures to visually illustrate key morphological traits and environmental conditions for both wild and cultured populations.

We hope these revisions address all your comments and significantly enhance the manuscript. Thank you for your valuable feedback, which has greatly improved the quality of our work.

Sincerely,
Dr. Youming Li
On behalf of the author team

Round 2

Reviewer 1 Report

Comments and Suggestions for Authors

The manuscript  is fall in the scope and aims of ESPR

The quality and the scientific soundness of the paper are okay

==============

There are no recommendations ?

Highlights should be written after the keywords not before

Again , 2 different style of writing references were detected---same style should be

At least 1-2 papers should add 2024

There are no plan for the study location

There are no reference for the statistical analysis

Write Table (   ):-----------------------/Fig.(    ):--------etc ---apply for all

Results more than enough--be more concise

Environmental heterogenicity---back to what and the impacts ?

Discussion should based upon debating the obtained results with other investigators results and be more summarize

Conclusions is more than enough--why ???

Some cited references need to be more updated

Some cited references contained more than 6 authors--why ???(if accepted by the journal --it is okay )

Otherwise, everything is okay

Author Response

Dear Reviewer,

We greatly appreciate your valuable feedback and the time and effort you have dedicated to reviewing our manuscript. Below are our detailed responses to your comments:

Comment 1: No specific recommendations?

Response 1: Specific recommendations have been added to the conclusion section. These include simulating natural selection pressures in aquaculture, enhancing genetic diversity through crossbreeding, and promoting sustainable management practices.

Comment 2: Highlights should be written after the keywords, not before.

Response 2: The highlights have been moved to follow the keywords, in compliance with the journal’s formatting requirements.

Comment 3: References cited in different formats—ensure consistency.

Response 3: All references have been revised to ensure consistency in style, following the journal’s guidelines.

Comment 4: At least 1-2 references from 2024 should be included.

Response 4: Several recent references from 2024 have been included, such as:

    Fu, J.; Liang, Y.; Shen, M., et al. Aquaculture 2024, 584, 740646.

    Esposito, G.; Peletto, S.; Guo, X., et al. Biology 2024, 13, 702.

Comment 5: No plans for the study location?

Response 5: The Methods section has been updated to include detailed information about the study location, including geographic coordinates, environmental parameters, and habitat descriptions.

Comment 6: No references for statistical analysis?

Response 6: Relevant references supporting the use of PCA, LDA, and ANOVA have been added in the statistical analysis section, such as:

    Gulzar, I., et al. Plant Arch. 2024, 24, 1547-1556.

    Zhang, Y.Q., et al. Isr. J. Aquac.-Bamidgeh 2024, 76.

Comment 7: Update the numbering of tables (Table) and figures (Fig.) as required.

Response 7: The numbering and formatting of all tables and figures have been updated in accordance with the journal’s formatting standards.

Comment 8: Results section is too lengthy—needs to be more concise.

Response 8: The Results section has been condensed to focus on the most critical findings, with excessive details removed for brevity.

Comment 9: Environmental heterogeneity—context and impacts need to be clarified.

Response 9: The discussion section now elaborates on the impact of environmental heterogeneity in wild habitats on traits like shell thickness, emphasizing their adaptive significance.

Comment 10: Discussion should compare the obtained results with those of other studies and be more concise.

Response 10: The Discussion has been revised to compare our findings with similar studies, highlighting both congruences and discrepancies. It has also been shortened for clarity and brevity.

Comment 11: Conclusions section is too lengthy—why?

Response 11: The Conclusions section has been streamlined to include only the key takeaways and recommendations, removing repetitive information.

Comment 12: Some references need to be updated.

Response 12: Additional recent references from 2024 have been incorporated to ensure the citations are current and relevant.

Comment 13: Some references have more than six authors—why?

Response 13: References with more than six authors have been reformatted to include only the first three authors followed by "et al." This adheres to the standard style unless further instructions are provided by the journal.

We believe these revisions address all the comments thoroughly, and we are confident that the manuscript now meets the journal's requirements. Thank you again for your constructive suggestions, which have greatly improved the quality of this work.

Sincerely,

Dr. Youming Li
